# Mast Cells and Skin and Breast Cancers: A Complicated and Microenvironment-Dependent Role

**DOI:** 10.3390/cells10050986

**Published:** 2021-04-23

**Authors:** Mark R. Hanes, Carman A. Giacomantonio, Jean S. Marshall

**Affiliations:** 1Department of Pathology, Dalhousie University, Halifax, NS B3H 4R2, Canada; Mark.Hanes@Dal.ca (M.R.H.); Carman.Giacomantonio@Dal.ca (C.A.G.); 2Department of Surgery, Dalhousie University, Halifax, NS B3H 2Y9, Canada; 3Department of Microbiology and Immunology, Dalhousie University, Halifax, NS B3H 4R2, Canada

**Keywords:** mast cells, tissue microenvironments, skin and breast cancers, model systems, therapeutic targets

## Abstract

Mast cells are important sentinel cells in host defense against infection and major effector cells in allergic disease. The role of these cells in cancer settings has been widely debated. The diverse range of mast cell functions in both immunity and tissue remodeling events, such as angiogenesis, provides multiple opportunities for mast cells to modify the tumor microenvironment. In this review, we consider both skin and breast cancer settings to address the controversy surrounding the importance of mast cells in the host response to tumors. We specifically address the key mediators produced by mast cells which impact tumor development. The role of environmental challenges in modifying mast cell responses and opportunities to modify mast cell responses to enhance anti-tumor immunity are also considered. While the mast cell’s role in many cancer contexts is complicated and poorly understood, the activities of these tissue resident and radioresistant cells can provide important opportunities to enhance anti-cancer responses and limit cancer development.

## 1. Introduction

Mast cells are myeloid cells characterized by their sensitivity to IgE-dependent stimulation and unique cytoplasmic granule contents such as histamine, tryptases, chymases, and carboxypeptidase A3. Mast cells intricately associate with vasculature and nerves in the major tissue types of most vertebrates and are also found in the peritoneal and thoracic cavities of many species. They are frequent at sites that interface with the external environment such as the skin, respiratory and gastrointestinal tracts. As front-line cells at the host-environment interfaces, mast cells promote host defense against pathogens by facilitating the initiation of appropriate immune responses [1,2]. Mast cells are renowned contributors to hypersensitivity reactions, particularly type-I hypersensitivity, through their release of histamine, and proteases, as well as producers of potent lipid mediators upon cross-linking of their Fc-receptors via IgE and antigen. Mast cells have also been suggested to facilitate the development of type-IV hypersensitivity reactions under some conditions. Notably, mast cells participate in tissue remodeling events and are inherently pro-angiogenic, promoting vessel formation through both constitutive and immunologically mediated release of angiogenic substances such as VEGF-A, CXCL16, Endothelin-1, GM-CSF, CXCL8, and CCL2 [3,4,5]. The location of mast cells at epithelial sites where tumor development often occurs, their ability to promote effective immune responses, and role in angiogenesis had led to an increased interest in the mast cell’s role in tumor development, tumor surveillance, and immune therapies. Accumulating data suggest significant and diverse roles for mast cells in tumor biology [6,7] and opportunities to potentially harness their functions therapeutically. In this review, we explore current understanding of the roles for mast cells in shaping common cutaneous and mammary cancers. We highlight the inherent pro-tumorigenic activities of mast cells in these contexts and how their actions can be modified by microenvironmental factors ranging from ultraviolet (UV) irradiation and microbiota to allergens and hypoxia. Our ability to target local mast cells therapeutically will also be evaluated in the context of our current understanding of mast cells as immune sentinel cells.

## 2. The Mast Cell—Tumor Controversy

The contribution of mast cells to tumor development and progression has proven to be a controversial area of research. On the clinical front, investigations of the relationship between allergic disease and various types of cancer as well as evaluation of mast cell location and density have yielded conflicting information [8]. Clinical studies have suggested a link between elevated IgE or the presence of allergic disease and reduced development of melanoma, breast cancer, and some types of brain tumor [9,10], while in other cancer settings this link has not been observed [11]. Interest in the field of AllergoOncology [12,13,14] has helped to bring together important information in this area. The potential for IgE-based cancer therapies is being actively explored [12,13,15]. In several studies, increased mast cells at solid tumor sites have been noted, in keeping with their localization at sites of blood vessel development [16,17]. The ability of mast cells to promote angiogenesis is viewed as a key process in promoting tumor development [18,19]. However, elevated mast cells at tumor sites or within draining lymph nodes have also been connected with improved outcomes. Similarly, mast cell activation at tumor sites has been reported to have widely differing effects. The rapid growth of many experimental mouse tumor models, and subsequent high dependency on angiogenesis may account for some of these differences. However, there are multiple additional factors within the tumor environment, host and experimental systems which influence the mast cell’s role.

Much of our understanding of the impact of the presence or absence of mast cells in tumors is derived from studies in mice using mast cell deficiency models. Differences in the nature of mouse mast cell deficiency models, specifically the use of c-Kit (CD117) deficiency-based models versus more selective mast cell deficiency models, have sometimes yielded conflicting results [19,20,21]. For example, DMBA/TPA skin carcinogenesis is unaffected in mast cell-deficient mice with wild-type c-Kit expression (Cre-Master) but is modified in mice with defective c-Kit (*Kit^W/W-v^*) [22,23]. Mast cell-deficiency models differ considerably in non-mast cell-related immune defects that can confound results and the extent to which they reduce mature and immature mast cell numbers locally and systemically [24,25]. They also often differ in genetic background. While c-Kit dependent models are open to criticism, especially when used without appropriate reconstitution controls, it is likely that no current mast cell-deficiency model is perfect. The full body of evidence must be carefully considered.

A recent trend in the field has been to breed mast cell-deficient mice with strains engineered to develop spontaneous or oncogene-inducible tumors. This has fueled new discoveries that help map the role of mast cells in skin- and breast-derived cancers [20,26]. For example, in an HPV-induced squamous cell carcinoma (SCC) murine model with *Mcpt5-Cre^+^ R-DTA^+^* mast cell-deficiency, mast cell status did not alter tumor growth or angiogenesis, contradicting an earlier study that used a c-Kit-dependent mast cell-deficiency model [19,20]. However, many roles for mast cells have yet to be validated using more than one tumor model. Given the wide range of solid tumor types and microenvironments, and the broad range of mediator responses that can be elicited from mast cells we should not assume that mast cells necessarily have a similar role or roles in multiple tumor settings.

## 3. Mapping the Role of Mast Cells in Cutaneous Cancers

The role of mast cells, and their prognostic and diagnostic implications in cutaneous tumors remains unclear [6,27,28,29]. It has been suggested that tryptase^+^ mast cell density is greater in melanoma and basal cell carcinoma than benign nevi and adjacent normal skin, respectively [28,29]. Moreover, invasive melanomas harbor fewer tryptase^+^ mast cells than dysplastic nevi or in situ melanomas [30]. Intriguingly, recent reports implicate high mast cell density as a signature for improved survival for patients with melanoma [6,30,31]. Along these lines, high numbers of tryptase^+^ mast cells have been found in regressing melanoma deposits relative to adjacent normal skin [6]. Overall, it is likely that mast cells have a strong effect on shaping cutaneous lesions, and much remains to be examined in this area.

Recent pre-clinical evidence from c-Kit-independent models of mast cell-deficiency suggests that, in the absence of further activation, mast cells are predominantly bystanders in orthotopic cutaneous lesions, unless appropriately activated (Table 1) [20,22,31,32]. This contrasts previous reports which demonstrated that resident mast cells enhanced the development of transplantable cutaneous tumors (e.g., B16-F10 melanoma) in c-Kit-dependent models such as *B6.Cg-Kit^W-sh/W-sh^* (Wsh) and WBB6F1/J-*Kit^W^/Kit^W-v^*/J [19,33]. One possibility is that complex alterations in function and density of immune cells, other than mast cells, in mice with compromised c-Kit expression confound results [25]. In support of the concept that mast cells serve only as bystanders in melanomagenesis, Ghouse et al. [20] showed that absence of local mast cells did not affect inflammatory cell infiltrates including CD19^+^ B cells, CD3e^+^CD8a^+^ T cells, CD3e^+^CD4^+^ T cells, CD11c^+^ cells, and CD11b^+^F4/80^+^ cells in B16-F10 melanomas or *K14-HPV16^+^* SCCs of c-Kit-expressing *Mcpt5-Cre^+^* mice. Moreover, the absence of mast cells had minor bearing on the transcriptome of purified macrophages from B16-F10 lesions [20]. Though, some mast cell subsets remain in this model, which may contribute to systemic anti-tumor immune responses or angiogenic processes.

## 4. The Function of Mast Cells in Malignant Breast Tumors

Although normal mast cell density is considerably less in mammary tissues relative to those observed in the skin, a role for these cells has still been suggested in breast cancer. Several studies have reported links between disease progression and survival and mast cell density or residence. Tryptase^+^ mast cells are often observed peritumorally in early-stage breast cancers without evidence of degranulation [34]. Notably, high tryptase levels are found in sera of untreated breast cancer patients [35] suggesting significant mast cell activation or turnover. During progression of non-triple-negative breast cancers, tryptase^+^ mast cells may increase in numbers and change in distribution from largely non-tumor associated with a contiguous peritumoral location, or in some cases to being localized to primary breast tumor stroma [36,37,38]. Mast cells are also enriched in the tumor bed and invasive margin of late-stage breast cancers, particularly luminal subtypes [37]. This transition in mast cell location is also linked with more carcinoma-associated fibroblasts (α-SMA^+^) and a gradual decrease in CD34^+^ stromal cells, which suggests that mast cells may participate in stromal remodeling through degranulation products, such as tryptase. This is in addition to reports of an association between local mast cells and angiogenesis [39,40]. Reports of mast cell degranulation are conflicting and often difficult to assess in post-surgical tissue samples [38]. However, since multiple mast cell mediators and chemoattractants can be produced without degranulation, the degranulation status of cells does not define their involvement in tumor regulation.

Interaction between KIR2DL4^+^tryptase^+^ mast cells and patient HLA-G^+^ breast cancer cells is reported to be associated with lymph node or in-transit spread [41]. Conversely, some studies have suggested that increased mast cells within tumor stroma were associated with an improved disease outcome [42]. Some differences between reports may be the result of the subsets of subjects examined since it has been observed that mast cells were observed in the tumor stroma more frequently in those subjects with high hormone receptive tumors compared with those that lack such receptors [43]. Intriguingly, a mouse model which involved crossing C57BL/6 MMTV-PyMT mice with mast cell-deficient Wsh mice suggests mast cells may program the developmental circuitry of murine breast cancer cells to a luminal-like subtype by directly enhancing *ESR1* transcription. This finding was supported by examination of human breast tumors [26]. Such mast cell-breast cancer cell crosstalk has also been speculated in patients with late-stage, chemotherapy-resistant inflammatory breast cancer as mast cells often resides in close proximity (≤5 μm) to cytokeratin^+^ breast cancer cells [44]. In breast cancer, as in cutaneous human tumors, the local impact of mast cells on tissue remodelling and cell recruitment events, and the effect of mast cells on draining lymph nodes/systemic immunity need to both be carefully considered and may not have similar disease impacts.

Data from murine c-Kit-dependent models of mast cell-deficiency suggest that mast cells serve as promoters of breast cancer development in this context. In a spontaneous Wsh-MMTV-PyMT breast cancer model, mast cell absence delayed onset of cytokeratin 5^+^ basal-like breast cancers. A lower tumor growth rate, reduced angiogenesis, and spread was observed in mast cell deficient compared with wild-type mice [26,45]. Moreover, mast cells have been implicated in promoting 4T1 and PyMT metastasis to bone and lung in part via the SCF/c-Kit-axis [46]. Supporting this mechanism, antibody blockade of c-Kit reduced breast cancer dissemination in a murine model that recapitulates mammary tumor relapse post radiotherapy [47]. However, much remains to be learned mechanistically about how mast cells modify the breast tumor microenvironment and how local mast cells can be manipulated to alter disease course. Future studies using c-Kit-independent models of mast cell deficiency are warranted.

## 5. Mast Cell Key Mediators That Influence Cutaneous and Mammary Tumors

Mast cells can produce a plethora of mediators in response to exogenous or endogenous ligands via pattern recognition receptors, immunoglobulin receptor-mediated activation, G-protein-coupled receptor activation, cytokine receptors and other pathways. In some cases, this activation is associated with short term mast cell degranulation, and in others to selective mediator release over minutes or days. In other settings mast cells play a pivotal role as sentinel cells responding to signals related to tissue damage or the presence of pathogens. Some of these danger-associated molecular pattern (DAMP) and pathogen-associated molecular pattern (PAMP) signals are also present at sites of tumor growth due to local hypoxia and associated adenosine and free radicals, acidity, tissue necrosis, tumor cell mediator production, and other processes. Such mediator production from mast cells is frequently involved with selective recruitment of effector cells such as granulocytes, natural killer (NK) cells, and T cells. However, the production of pro-angiogenic factors such as VEGF-A isoforms [48,49] and VEGF-C, VEGF-D, and VEGF-F which have key roles in lymphangiogenesis [50,51,52] and lipids (i.e., LTB4, LTC4) [53] with or without degranulation are also features of responses involving mast cells. Some key mast cell mediators modulating tumor development are discussed below.

### 5.1. TNF Family Members

Possibly the most widely studied mast cell-associated cytokine is TNF. Mast cells can produce de novo and release stored free and granule-associated tumoricidal TNF upon activation via pattern recognition receptors or signalling through FcεRI (Figure 1) [54,55]. TNF, like many other mast cell-derived cytokines, can also be produced long term by degranulation independent mechanisms. In some situations, TNF from mast cells can be directly cytotoxic [56]. TNF has also been implicated in governing neutrophil infiltration of inflamed skin [57], mast cell-mediated dendritic cell (DC) mobilization [58,59,60], as well as the local recruitment of effector cells into mast cell rich sites [61]. TNF can also enhance mast cell-mediated proliferation of CD3^+^ T cells in culture [62]. The role of TNF as an inducer of other inflammatory cytokines such as IL-1β and IL-6, means that it can promote an inflammatory response locally and act in both an autocrine and paracrine manner to enhance systemic inflammation. In some cases, this could be a useful tool to enhance anti-tumor immunity or enhance tumor immunotherapy, but in other contexts such responses could promote harmful T_H_17 responses, angiogenic events, and metastasis.

In IL-6 rich zones, mast cells can modulate T_REGS_ via OX40L-OX40 engagement to alleviate their suppressive functions [63,64]. OX40L neutralization has been shown to decrease IgE/antigen-dependent mast cell-mediated activation of CD3^+^ T cells [65]. OX40L expressing mast cells also promote IFNγ release from NK cells in culture following exposure to agonists of TLRs 3, 4, and 9 [66]. Mast cells may also have the ability to express B cell-activating factor (BAFF) and TNF-related apoptosis inducing ligand (TRAIL) [67,68]. Exogenous BAFF slows B16-F10 melanoma growth in vivo and drives T_H_1 responses potentially further enhancing anti-tumor immune responses [69]. However, in the setting of breast cancer, BAFF and the related mediator, APRIL, may promote metastasis [70]. Similarly, TRAIL can selectively induce apoptosis of epithelial tumor cells such as melanoma expressing its death receptors 4 or 5, but is implicated in metastasis of cancers at other sites such as pancreatic [71,72]. Thus, the roles for these TNF family members in the tumor microenvironment are complex and the importance of mast cells as a source or responding cell type has yet to be addressed.

### 5.2. IL-6

Mast cells are unusually potent sources of IL-6. In response to some stimuli, such as LPS, some mast cells can produce 10-fold more IL-6 per cell than similarly activated macrophages [73]. Elevated levels of circulating IL-6 is often viewed as a negative prognostic indicator in multiple cancers [74,75]. Such IL-6 can be the result of tumor cell IL-6 expression or production by a number of other cell types in response to DAMPs, hypoxia, or inflammatory mediators. IL-6 signaling enhances STAT-3 dependent pro-oncogenic pathways [76,77]. However, local mast cell-mediated IL-6 production during tumor development can act to enhance the process of DC mobilization and enable the transmigration of effector cells from the blood stream into local tissues [33,78]. IL-6 is a pleotropic cytokine with complex and varying impacts on tumor growth and anti-tumor immunity. Local production at tumor sites from mast cells is likely to have significantly different effects than systemic increases in cytokine levels.

### 5.3. IL-13

When activated via IgE/antigen or by certain DAMPs, such as IL-33, mast cells are an excellent source of IL-13 [79]. This cytokine has a complex role in tumor immunology. In a mouse model of human ovarian cancer, IL-13 was reported to promote metastatic activities through the high-affinity IL-13 α2 receptor [80]. However, another study showed that induction of IL-13 via IL-33 injection delayed murine ID8 ovarian cancer progression [81]. Moreover, IL-13Rα2 is considered a tumor marker for cancers such as glioblastoma multiforme, melanoma, and luminal breast cancer [82,83,84]. Exposure to high concentrations of IL-13 induced human lung-derived mast cell proliferation and enhanced histamine release [85]. Overall, IL-13 appears to have a strong effect on modulating tumor progression. Much remains to be discovered on the role of IL-13 in tumor beds including processes of macrophage polarization and efferocytosis.

### 5.4. IL-17A

The IL-17 cytokine family is comprised of six distinct isoforms IL-17A-IL-17F. The ability of mast cells to produce IL-17 isoforms is controversial [86]. Regardless, mast cells might capture free IL-17A and serve as important reservoirs or vehicles for IL-17A in tumor beds [86,87]. Therefore, it is possible that mast cells prolong local IL-17A release, which links mast cells to the immunopathogenic functions of IL-17A in cancers [86,88]. Indeed, primary growth and metastasis of B16-F10 melanoma and 4T1 breast cancer was reduced in mice depleted of IL-17A by host knockout or antibody neutralization, respectively [89,90]. Similarly, tumorigenesis of DMBA/TPA skin carcinogenesis was alleviated in IL-17A-null mice [91]. Along these lines, high expression of IL-17A-expressing cells is a strong indicator of disease progression in patients with head and neck cancers and of poor prognosis in those with breast cancer [92,93]. It is unclear if mast cells can capture other IL-17 isoforms and the impacts of these in tumor microenvironments. Though, based on sequence homology IL-17F may share many functions with IL-17A in hosts with skin or breast cancers [94,95].

### 5.5. Mast Cell-Derived Chemokines

A variety of chemokines are produced by mast cells. In some cases, microenvironmental factors, as discussed below, DAMPs from stressed cells or local microorganisms will lead to the activation of tumor-associated mast cells. Chemokines, such as CCL3, CCL4, and CCL5 can promote the recruitment of T cells to the tumor site [96]. Others such as CXCL8 contribute to the recruitment of NK cell and neutrophils [96]. Such mast cell-derived chemokine signatures have dual outcomes in providing enhanced inflammation which may promote tumor growth through angiogenesis and providing recruitment signals to anti-tumor effector cells.

### 5.6. Histamine

Histamine is regarded as essential for tissue neovascularization and is produced by mast cells and basophils. Indeed, at a tissue level, mice deficient in histidine decarboxylase (*Hdc*^−/−^), the enzyme that catalyzes histamine formation, have reduced angiogenesis and delayed wound healing compared with wild-type mice [97]. *Hdc*^−/−^ mice are more susceptible to skin carcinogenesis and have a greater disease burden in melanoma models [98]. It is also known that histamine can promote the mobilization of local DCs to the draining lymph node [99]. The clinical use of H1-receptor antagonists, desloratadine and loratadine, are associated with improved survival in patients with melanoma [100]. In contrast, H2 receptor antagonists, but not H1 blockade, have been shown to inhibit breast tumor growth in multiple mouse models. This may be related to inhibition of myeloid-derived suppressor cell (MDSC) development [101] and involve modulation of B cell responses in some models [102,103]. Unfortunately, data are currently not available on the use of either H1 or H2 blockade in larger cohort studies examining cancer incidence or progression, since the use of these “over the counter” agents is often unrecorded.

### 5.7. Tryptase

Historically, mast cell tryptase is renowned for its pro-tumorigenic role via enhancement of angiogenesis [104]. Namely for inducing endothelial cell proliferation and tube formation, and breakdown of connective tissue matrix to facilitate neovascularization. Illustrated in Figure 2 are tryptase^+^c-Kit^+^ mast cells in a skin biopsy from a patient with melanoma resistant to immunotherapy. Tryptase is secreted well beyond the cell boundaries as defined by positive mast cell membrane c-Kit staining. This suggests ongoing tryptase secretion at this tumor site, a pattern that is also observed in many other solid tumors.

A recent study has demonstrated that mast cell tryptase drives nuclear remodeling in murine and human melanoma cells to inhibit the proliferation of melanoma cells and alter their expression of antigens [105]. Another novel study by Grujic and colleagues [106] showed that mice deficient in tryptase (Mcpt6^−/−^) develop larger B16-F10 lesions than control mice. Moreover, absence of tryptase, chymase, and carboxypeptidase A3 resulted in increased melanoma cell colonization of lungs in an intravenous B16-F10 model [107]. These data suggest that tryptase production may allow certain subtypes of melanoma to remain in a senescence-like state rather than actively dividing. It is unclear whether this novel function for tryptase is also relevant to other tumor types in mast cell rich tissues.

## 6. Environmental and Physiological Pathways That Differentially Program Mast Cells

Mast cell differentiation and functions are regulated by their tissue microenvironment and differ between anatomical locations. In the context of solid tumors, tumor-derived cytokines and other mediators can affect mast cell function and recruitment. These include SCF, IL-6 and others, the profile of which are highly tumor-type dependent and not the focus of this discussion. Other stimuli add to this complexity for example mast cells resident to skin areas of high sun-exposure may promote inflammatory immune reactions [108]. Conversely, mast cells in breast tissue are largely protected from UV irradiation, but may be exposed to alternate bacterial stimuli and tissue changes associated with milk production. Here, we outline how some of these physiological challenges may influence the activities of mast cells important to tumor development and development of effective cancer immunity.

### 6.1. Ultraviolet Irradiation

Mast cells are thought to be effector cells during UV ray-mediated inflammatory responses involving the release of DAMPs such as HMGB1. However, few studies have examined this issue in a cancer context. HMGB1 prompts PAR-2- and TLR4-mediated local neutrophilic responses characterized by IL-6, CXCL8, and TNF [109,110,111]. Mast cells likely add to this pool of CXCL8, IL-6 and TNF in response to TLR activation [112,113]. Inflammation is associated with cancer development in several tissue settings [114]. In the skin, elevated TNF is involved with SCC development [115]. Moreover, TLR4-ligands can also directly promote tumorigenesis in melanoma [116,117]. Specifically, LPS-induced STAT3-signaling, as a result of TLR4 activation, elevates B16-F10 melanoma growth, angiogenesis, and invasion in vitro and in vivo [116]. Overall, UV-induced cell damage might foster mast cells to contribute to early-stage cutaneous T_H_1 cytokine profiles [118,119]. Although controversial, there is some evidence that at later time points following UVB irradiation mast cells might be enhanced in numbers through CXCL5-dependent migration. This would be predicted to both enhance local capacity to recruit anti-cancer effector cell populations and to promote further inflammatory cytokine responses [118,119,120,121]. In addition, UVB triggers enzymatic assembly of 1α,25(OH)_2_D_3_, an active metabolite of vitamin D_3_, which dampens cutaneous IgE-mediated mast cell responses. It is unclear whether this process modulates the response of mast cells or other effector cells to local tumor development. Similarly, more studies are needed to define the role of other UVB by-products such as fibronectin-binding protein and *cis*-UCA on the biology of tumor-resident mast cells [122,123,124,125].

Moreover, UVB irradiation and lipid mediators, including platelet-activating factor (PAF), upregulate IL-33 expression on epithelial cells that can in turn act to induce T_H_2 cytokine production by mast cells [126,127]. IL-33 also supports human cutaneous mast cell population expansion while having limited impact on degranulation responses [128]. Interestingly, IL-33 is highly expressed on immune resistant, UV-derived epidermal SCCs, such as 13-1, but not SCCs susceptible to immune rejection (e.g., LK-2) [126].

Cutaneous inflammation from UVB may also modulate adaptive responses through impacts on mast cells. UVB-generated PAF binds to G-protein coupled PAF receptors on mast cells to enhance histamine release and generation of PGE_2_ [129]. Importantly, PAF has been reported to promote mast cell migration to lymph nodes, particularly B cell zones, by upregulating CXCR4 on mast cells and its ligand, CXCL12, in draining lymph nodes [130,131]. DC migration to nodes might also be enhanced due to UV-mediated events. Overall, UV irradiation may facilitate primary immune responses by driving APCs into tumor-draining lymph nodes.

Dermal mast cells can alter T and B cell responses by modulating classical DC subsets. UVB-exposed murine dermal CD11c^+^Langerin^+^ Langerhans cells navigate to lymph nodes to inhibit type-IV hypersensitivity through UVR-Treg induction [132]. This may further support an immunosuppressed environment at chronically UV exposed sites at higher risk for tumor development.

### 6.2. Microbiota

The microbiome of mammals is tissue site, species- and sex-specific [133,134,135,136]. Mast cells may be differentially influenced by the unique microbiomes of tissues such as the breast and skin. In the context of disrupted tissue architecture and immunosuppressive cancer treatments mast cell interaction with both microbes and microbial products may be enhanced.

The microbiome of malignant breast lesions is greater and more diverse than patient-matched normal adjacent tissue, normal breast samples from healthy subjects, and other tumor types [137]. Tumors of the breast have been shown to harbor nearly 175 bacterial genera, whereas melanomas generally fall below 25 genera. Melanoma microbiomes are enriched in Gram-positive species that are known TLR2 activators. In contrast to high lipoteichoic acid (LTA) expressing bacteria levels in melanoma, LTA levels are low to absent in breast tumors [137]. This suggests that the microbial microenvironment of breast tumors may be enriched with Gram-negative bacteria, the latter being activators of TLR4-signalling. Gram-positive and -negative bacterial products both stimulate mast cell production of TNF, IL-6, GM-CSF, IL-1 and IL-13 in mice. However, in human mast cells, TLR4 expression is inconsistent and may be limited to certain mast cell subtypes or IL-4 enriched microenvironments [138,139,140]. Broad-spectrum depletion of mainly Gram-negative bacteria with aminoglycosides has been shown to enhance breast tumor growth in a HER-2/*neu* spontaneous mammary carcinoma model [141]. This could suggest that innate immune stimulation via the microbiome modifies tumor growth, although the role of the mast cell in this process is poorly defined. Interestingly, a recent report demonstrated that the skin microbiome protected specific pathogen-free mice against UVB ray-associated type-IV hypersensitivity immunosuppression relative to germ-free mice [142]. The microbiome could be a contributing factor for the wide diversity in findings regarding the role of mast cells in such hypersensitivity responses.

### 6.3. Hypoxia

Solid tumors including those of skin or breast origin are often associated with local tissue hypoxia resulting from dysfunctional vasculature or increased metabolic demand. This can contribute to the development of areas of central necrosis within tumors. Hypoxic environments are highly immunosuppressive due in part to low pH and elevated concentrations of ROS, adenosine, and nitric oxide [143]. Interestingly, mast cells are frequently localized in hypoxic regions of tissues such as melanoma and associated with the active angiogenic processes initiated in these microenvironments [5,144].

A moderate level of hypoxia does not induce significant cell death of bone marrow-derived mast cells, nor does it inhibit their degranulation [144]. Instead, hypoxic conditions drive HIF1α and subsequent ROS production in exposed mast cells as seen in vitro studies [144]. Significantly, hypoxia transiently elevates mRNA levels of VEGF-A and TNF above baseline and stably promotes CCL2 protein production, the latter being dependent on ROS and LVDCC [144]. As determined by confocal microscopy, there is high overlap between tryptase^+^ mast cells and hypoxic regions (pimonidazole^+^ cells) in murine melanomas including B16-F1 and -F10, along with overlap with CCL2 expression in the former [144]. These data suggest that cutaneous mast cells could contribute to tumor growth and local inflammation by production of ROS, and VEGF-A and through recruitment of CCR2^+^ MDSC and macrophage populations.

In breast cancer an alternate mechanism has been suggested whereby resident mast cells may play a role in reducing hypoxia. In a 4T1 murine breast cancer model, stabilization of mast cells with disodium cromoglycate increased intratumoral pimonidazole expression which indicates the level of hypoxia and blood clotting [34]. This may reflect a reduction in mast cell contribution to angiogenesis through release of preformed mediators such as VEGF and tryptase. These studies demonstrate how the impact of mast cell function can change depending on the specific tissue and tumor microenvironment.

A key metabolic by-product that modulates mast cells at hypoxic sites is adenosine. Hypoxic or ischemic tumors feature elevated levels of adenosine from the catabolism of ATP (released from dying and stressed cells) involving the ectonucleotidases CD39 and CD73 [145]. High CD73 expression in tumors has been reported to foster therapy resistance and poor prognosis in specific skin and breast cancers [146,147,148]. Interestingly, rat and murine-derived mast cells are reported to secrete adenosine in response to calcium ionophore stimulation [149]. Though, it is unclear if mast cells contribute to the pool of adenosine present in tumors. Mast cells have been reported to express all adenosine receptors (A_2A_, A_2B_, and A_3_, and A_1_) across multiple species under specific activation states [150,151]. Adenosine appears to enhance the angiogenic activities of mast cells. A_2B_-signaling in mast cells induced VEGF and CXCL8 secretion, whereas A_3_-stimulation increased angiopoietin-2 protein levels [152]. A role for A3-signaling in mast cell degranulation has also been documented [153]. Moreover, medium from mast cells activated by A_2B_- and A_3_-agonists proved superior in driving capillary tube formation of HUVECs than either agonist alone [152]. Additional tumor-promoting roles for adenosine include suppression of anti-tumor T cells and NK cells while supporting T_REG_ and MDSC function and immunosuppression predominately through A_2A_-signaling [154,155]. Overall, however, very little is understood about the effects of adenosine on mast cell functions within tumors.

These few examples demonstrate that the presence of mast cells and their responses to environmental and microenvironmental factors can have multiple downstream impacts. Other physiological cues not discussed but also capable of influencing mast cell function include temperature, oxygen, and pH levels. The cytokines produced by other immune cells, structural cells, and tumor cells in response to such stimuli may also govern mast cell activities. Mast cells express receptors for multiple mediators such as TNF, IL-10 and VEGF allowing for complex crosstalk in tissue networks.

## 7. Therapeutically Targeted Mast Cells Drive Anti-Tumor Responses

Mast cell activation has not been systematically examined as a therapeutic approach to cancer therapy in clinical studies. Several approaches which involve activation of local mast cell populations in lung and skin tumor models have been examined in animal models. In most cases, the aim is to enhance effective anti-tumor immunity and/or selectively kill cancer cells.

A key feature of the long-term resident tissue mast cells found in both skin and breast tissue is that they have an extensive tissue half-life and do not divide substantially in situ. This contributes to their resistance to negative impacts from standard therapies such as irradiation and multiple cytotoxic drugs. Mast cells remain viable and retain the ability to both degranulate and produce a host of newly formed mediators such as cytokines and chemokines following irradiation [156,157,158]. In this context, mast cells are excellent targets for appropriate activation by adjuvant immunotherapies or via appropriate antibodies, some examples of which are discussed below.

### 7.1. Mast Cells and IgE in AllergoOncology

Mast cells are well known to express high affinity receptors for IgE which are critical for their role in host defense against nematodes and in allergic disease. They also have receptors for other immunoglobulin subclasses and free light chains (FLC), although expression of these varies with species. Mast cells may engage in crosstalk with immunoglobulin-expressing tumors as seen with respiratory epithelial cells [159] or EL4 lymphoma [160,161] to either support anti-tumor host responses or tumor escape.

IgE antibodies against a wide variety of cancer types have been observed clinically and multiple IgE monoclonals have been employed in pre-clinical models [12,14,162,163]. They have been demonstrated to facilitate ADCC by several cell types including monocytes and eosinophils, and to induce mast cell degranulation. The latter process is associated with the production of several chemokines with the ability to recruit anti-tumor effector cells, in addition to more pro-tumorigenic and pro-angiogenic mast cell responses. A number of clinical and epidemiological studies have demonstrated a link between IgE antibodies and reduced incidence of breast cancer, melanoma, and glioma [14,164], although in several other cancers such a link has not been observed. Evidence has also accumulated suggesting a protective function for IgE/antigen and mast cells in pre-clinical models. For example, using mice with high IgE-titres (KN1), Nigro et al., 2016 [165] demonstrated that host IgE protected against N2C breast tumor establishment and had reduced growth of TS/A breast tumors relative to controls. Importantly, the anti-tumor activity of IgE was reliant on the high-affinity IgE-receptor, FcεRIα [165]. Additionally, trastuzumab IgE, an antibody raised against HER2/*neu* (a receptor common to HER2^+^ breast cancer) provided equal if not added cytotoxic benefit compared to the IgG1 form of the same antibody [166]. Anti-HER2/*neu* IgE was reported to drive mast cell degranulation and generation of newly synthesized mediators including GM-CSF and TNF [166,167]. Of note, in vitro, human mast cells sensitized with anti-HER2/*neu* induced TNF-mediated apoptosis of SK-BR-3 breast cancer cells [167]. Furthermore, in a transgenic hFcεRIα murine model, 4T1 breast cancer was controlled with locally produced anti-hMUC1 IgE [168]. IgE/antigen-activated mast cells may release mediators such as CCL2, CCL3, CCL7, and CCL8 among others (Figure 1) that facilitate inflammatory innate reactions and subsequently promote adaptive immunity against breast tumors.

It must be mentioned that IgE-mediated degranulation may also promote angiogenic function of mast cells. Exogeneous IgE increased B16-F10 melanoma growth and the density of blood vessels in mast cell-sufficient but not Wsh mast cell-deficient mice [169]. Reconstitution of Wsh mice with wild type but not Fyn kinase^−/−^ BMMCs rescued this feature of IgE treatment. Fyn kinase regulates the de novo synthesis of VEGF in IgE-activated mast cells [169].

The potential protective attributes of IgE in cancer therapy have yet to be exploited therapeutically despite supportive pre-clinical evidence [166,170,171]. This may be related to concerns regarding the potential for inducing anaphylaxis. High levels of galectin 3 and galectin 9 which can bind IgE may also modulate efficacy in select tumor microenvironments [172,173]. Most clinical antibody-based modulators including checkpoint blockades PD-1/PD-L1 and -CTLA-4 are of IgG isotypes. MOv18 is the only IgE-based antibody currently undergoing clinical investigation for therapy of late-stage solid tumors (NCT02546921) and targets the alpha folate receptor expressed highly on several cancer types. MOv18 IgE has been examined in several pre-clinical trials [166,170]. IgE proved superior to other antibody isotypes in managing tumor growth and prolonging survival without anaphylaxis being observed. This could suggest a mast cell or basophil induced anti-tumor impact.

IgE and mast cells may also have a role in immune surveillance. A strong inverse association has been reported between total and allergen-specific serum levels of IgE and the risk of melanoma in the overall population, and with risk of breast and gynecological cancers in women [9]. It is tempting to speculate that tumor laden mast cells contribute to tumor clearance after activation by IgE/antigen. However, high IgE levels are associated with multiple other changes in immune function which may explain the differences observed.

Free light chains have also been suggested to have a role in modifying anti-tumor immunity. Kappa and lambda FLC protein expression is positively correlated with increased breast tumor size, tumor grade, clinical stage, and vascular invasion. It has been reported that signaling through a FLC-receptor may modify mast cell responses to tumors and enhance tumor growth. Lambda FLC expression is mainly associated with stromal inflammatory cells in breast tumors [174]. Studies using the Wsh mast cell deficient mouse model suggested a role for mast cells in this process [174]. However, these studies remain to be confirmed in non-c-Kit-dependent models of mast cell deficiency.

### 7.2. TLR-Signaling

Mast cells can express multiple TLRs such as TLRs 1, 2, 3, 4, 6 and 9. These can be activated by exogenous ligands such as LPS (TLR4) or Pam3CSK4 (TLR2) as well as by local DAMPs such as HMGB1 or CpG motif rich DNA. In mice, mast cells promote effector T cell and NK cell recruitment and subsequent responses that combat tumor growth, when activated by TLR agonists. In earlier studies of the potential impact of activated mast cells on tumor growth Oldford et al., 2010 [33] showed that mast cells exposed to the TLR2 agonist, Pam_3_CSK_4_, facilitate the recruitment of CD3^+^ T cells in addition to NK1.1^+^ NK cells to B16-F10 lesions in a CCL3-dependent manner. This process was dependent on mast cell IL-6 production [33]. Similarly, in response to TLR4 stimulation by LPS, tumor-resident mast cells release CXCL10 and can recruit antigen specific T cells to B16-F10-OVA lesions [6]. This mechanism may be especially important in the context of a systemic LPS signature release in response to checkpoint inhibitor treatment in cancer patients. Mouse studies, examining the mechanism of such responses revealed that the numbers of tumor-infiltrating T cells were significantly lower in the absence of mast cells. Cutaneous tumor control was also lost when mast cell-deficient mice were reconstituted with *Cxcl10*-deficient mast cells. This highlights a role for mast cell-derived CXCL10 in T cell recruitment to melanomas [6].

Mast cell activation via a variety of mechanisms can also induce the maturation and mobilization of local DC populations. In a peptidoglycan-driven model, we showed that Langerhan’s cell migration to draining lymph nodes was dependent on mast cell presence [58,99]. DCs also home to draining lymph nodes in response to IgE/antigen-activation of mast cells via a histamine-dependent mechanism [99]. Other DC subsets, including those strongly implicated in effective anti-tumor immunity also migrate to draining nodes as a result of mast cell activation using pathways with distinct mast cell mediator dependencies [78].

The effects of mast cell activation are not limited to TLR2 and TLR4 mediated events. For example, imiquimod, a TLR7 agonist, triggers CCL2 secretion from B16-F10-resident mast cells to mobilize anti-tumor CD8α^+^ plasmacytoid DCs to clear B16-F10-melanomas in a IFNAR1-dependent manner [175]. Topical imiquimod may also increase mast cells in treated actinic keratosis sites [176].

### 7.3. Virus Mediated Activation

The use of oncolytic viruses and viral associated activators in cancer therapy have been explored for many years [177,178,179]. Early in vitro studies using human cord blood-derived mast cells showed that an oncolytic mammalian reovirus (serotype 3 Dearing) can induce recruitment of CD56^+^ T cells through CCL3, CCL4, and CCL5 release and NK cells via CXCL8 [180,181]. More recently, we have demonstrated that the products of virus infected mast cells can enhance the activation and tumor cell killing ability of NK cells via a type-I IFN related mechanism [182,183]. Indeed, mast cells are an unusually potent source of type-I and type-III IFNs following infection with several viruses, allowing them to locally activate multiple potential anti-tumor mechanisms and subsequent IFN regulated responses. In a dengue virus model, cutaneous viral-induced inflammation induced CXCL10 expression by mast cells. This might be attributed to mast cell exposure to IFNs [184], or by a more direct result of viral infection. In a mouse Dengue virus model system, mast cells recruited CD69 expressing (activated) CD3^+^CD8^+^ T cells, CD3^+^NK1.1^+^ NKT cells, and CD3^+^γδTCR^+^ γδ T cells to infected skin and draining lymph nodes [185]. Ongoing clinical studies using attenuated dengue virus to treat advanced-stage melanomas may provide insights into the utility of this approach to enhance effective anti-melanoma immunity (e.g., NCT03989895). Other viral therapeutics such as the reovirus-based “reolysin” may exploit the mast cell response to reovirus, as described above, to mobilize effective NK cell and T cell responses.

## 8. Conclusions and the Future Landscape

Despite substantial progress, the goal of reproducing human disease and anti-tumor immune responses in murine models remains elusive. Although we have a good understanding of how mast cells may both promote or limit tumor growth, their true roles in most human cancers remains poorly defined. Tumor models are both phenotypically and functionally heterogeneric, with unique cellular networks, abilities to spread, and angiogenic rates [186]. With this context in mind, it is important not to label mast cells with specific functions based on findings using a single tumor model. The use of transient mast cell knock-out mice such as *Mcpt5-Cre; iDTR* may serve as one useful tool in unraveling the role of mast cells at distinct stages of tumor development [187]. However, multiple complementary models are needed to ensure the validity of such mouse studies.

Environmental factors such as UV irradiation and microbial colonization, and local hypoxia have substantial impacts on mast cell populations and function. Similarly, the cytokine microenvironment and the presence of mast cell activators such as IgE/antigen and multiple pathogen products modify their functions. Mast cells remain powerful, tissue resident immune sentinel cells that are often enriched at solid tumor sites. In most solid tumors they can relieve local immunosuppression and enhance tumor cell killing when appropriately activated. In many cancer settings, mast cells also have the potential to induce effector cell recruitment, increase the opportunity for effective DC mobilization and subsequently promote induction of acquired immunity. The ability of mast cells to produce mediators that can promote tumor development, has been demonstrated in several model systems and also needs to be carefully considered. The ability of mast cells to promote tumor angiogenesis is a particular concern for rapidly growing solid tumors where blood supply limitations can be most critical. Mouse models of disease, especially rapidly developing transplantable tumor models, may not reflect the balance of positive and negative effects of mast cells in human disease accurately. However, newer models of disease and improved tools such as single-cell sequencing, high-parameter cell phenotyping, metabolic profiling and imaging for enhanced detailing of human tissues should further advance understanding of the role of mast cells in skin and breast cancers and other diseases. Harnessing these techniques to study mast cell–cellular networks within skin and breast cancers is likely to uncover novel functions of mast cells and provide a blueprint for selectively altering mast cell functions for therapeutic benefit.

## Figures and Tables

**Figure 1 cells-10-00986-f001:**
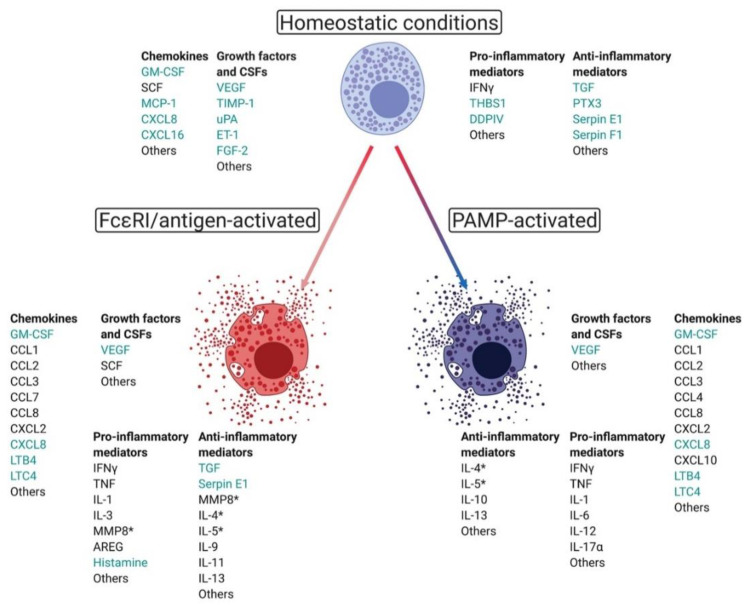
Human mast cell circuitry is programmed by host and environmental factors and is largely conserved in mice. Unstimulated mast cells are inherently pro-tumorigenic. Mast cells in homeostatic states release angiogenic mediators (teal font) such as VEGF, uPA, and MCP-1 that are able to promote tumor survival and progression. Mast cells activated by FcεRI/antigen-stimulation often adopt a less pro-tumorigenic phenotype and release a series of chemokines (CCL2, CCL3, CCL7, CCL8) that attract members of the mononuclear phagocyte system to tumor sites, where they amplify inflammatory responses. Additionally, PAMPs can program mast cells into anti-tumor effector cells capable of lysing tumors through TNF and IFNγ release and recruiting CD8^+^ T cells and NK cells to sites of tumors via mediators such as CXCL10 and CCL3, respectively. Moreover, effector mast cells release less angiogenic and pro-inflammatory mediators. Asterisks indicate both pro- and anti-inflammatory functions. Figure prepared in BioRender.

**Figure 2 cells-10-00986-f002:**
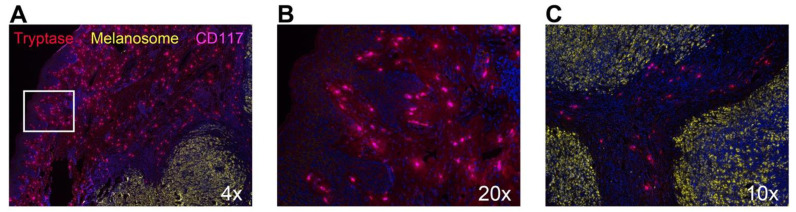
Mast cells resident in the epithelium and stroma of a patient with biotherapy-resistant cutaneous melanoma. (**A**–**C**) Fixed and paraffin-embedded melanoma sections were labeled against tryptase (red), CD117 (c-Kit; pink), melanosome (yellow), and DAPI (blue) using Opal reagents (Akoya Bioscience). Slides were scanned using the Akoya Biosciences Mantra 2^TM^ Quantitative Pathology Workstation at either 4× (**A**), 20× (**B**), or 10× (**C**) magnification. (**A**) Representative image illustrating the abundance of tryptase- and CD117-expressing mast cells surrounding melanosome-expressing biotherapy-resistant melanoma. (**B**) Zoom of mast cells in the boxed region in (**A**,**C**), Mast cells predominately populate the stroma of biotherapy-resistant melanoma.

**Table 1 cells-10-00986-t001:** Comparison of mast cell functions in mice with skin-derived tumors.

Cancer Type	Tumor Model	Mouse Strain	MC-Deficiency Model	MC-Modulator	Role	Publication
Melanoma	B16-F10 (S.c.)	C57BL/6	*Kit^W^/Kit^W-v^* (c-Kit-dependent)	None	Pro-tumor	Starkey et al., 1988 [4]
Melanoma	B16-F10 (S.c.)	C57BL/6	*Kit^W-sh^/Kit^W-sh^* (c-Kit-dependent)	None	Pro-tumor	Oldford et al., 2010 [33]
Melanoma	B16-F10 (S.c.)	C57BL/6	*Mcpt5-Cre^+^R-DTA^+^* (c-Kit-independent)	None	No contribution	Öhrvik et al., 2016 [32]
Melanoma	B16-F10 (I.d.)	C57BL/6	*Mcpt5-Cre^+^R-DTA^+^* (c-Kit-independent)	None	No contribution	Ghouse et al., 2018 [20]
Melanoma	Tg(GRM1)Epv (Spontaneous)	C57BL/6	Cre-Master (c-Kit-independent)	None	Anti-tumor	Stieglitz et al., 2019 [31]
Skin carcinogenesis	DMBA/TPA (Topical)	C57BL/6	Cre-Master (c-Kit-independent)	None	No contribution	Antsiferova et al., 2013 [22]
Skin carcinogenesis	DMBA/TPA (Topical)	WBB6F1	*Kit^W^/Kit^W-v^* (c-Kit-dependent)	None	Anti-tumor	Siebenhaar et al., 2014 [23]
Squamous cell carcinoma	K14-HPV16 (Spontaneous)	FVB/n	*Mcpt5-Cre^+^R-DTA^+^* (c-Kit-independent)	None	No contribution	Ghouse et al., 2018 [20]
Squamous cell carcinoma	K14-HPV16 (Spontaneous)	FVB/n	*Kit^W^/Kit^W-v^* (c-Kit-dependent)	None	Pro-tumor	Coussens et al., 1999 [19]

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
