# Peer review of "Mast Cells and Skin and Breast Cancers: A Complicated and Microenvironment-Dependent Role"

_cells, 2021, doi:10.3390/cells10050986_

Round 1

Reviewer 1 Report

This is an interesting and timely review on the role of mast cells in cancer focusing on their still highly complex role in skin and breast cancer. The review extensively discusses the published literature and describes their their complex and microenvironment-dependent role as well as the role of released compounds. My comments are only minor :

  • the authors extensively discuss the role of mast cells in the the tumor environment in the relevant tissues. However, little information is provided concerning their role in secondary lymphoid organs (see lines 160/161). In case there is some more information available this should be added.
  • concerning the role of proteases an important reference from the group of G. Pejler concerning the beneficial role of triple KO mice (chemise, tryptase CPA3) in melanoma is missing (PMID: 28212574).
  • Concerning IgE-mediated therapies there is some interesting older information published by Shiver and Henkart PA in Cell 1991 and 1992 indicating that RBL mast cells have some cytotoxic activity when transfected with perforin and Granzyme A, this may be worth to discuss.
  • line 66 some reference concerning mast cell promotion of angiogenesis is missing
  • line 414 a reference concerning adenosine receptors seems to be missing
  • line 192/223 typos FcepsilonRI 

Author Response

Reviewer 1:

This is an interesting and timely review on the role of mast cells in cancer focusing on their still highly complex role in skin and breast cancer. The review extensively discusses the published literature and describes their complex and microenvironment-dependent role as well as the role of released compounds.

Comment 1: The authors extensively discuss the role of mast cells in the tumor environment in the relevant tissues. However, little information is provided concerning their role in secondary lymphoid organs (see lines 160/161). In case there is some more information available this should be added.

Response: Thank-you for the suggestion. Mast cells are only found in very low numbers in secondary lymphoid tissues, which is likely the main reason for a deficit of literature on the role of the mast cell in this setting, even in a tumor context. We are not aware of substantial literature in this area to include.

Comment 2: Concerning the role of proteases an important reference from the group of G. Pejler concerning the beneficial role of triple KO mice (chemise, tryptase CPA3) in melanoma is missing (PMID: 28212574).

Response: The original paper by Grujic and colleagues (2017) highlighting the impacts of tryptase, chymase, and CPA3 deficiency on B16-F10 melanoma has been added (page 13). Thank you for this important suggestion.

Comment 3: Concerning IgE-mediated therapies there is some interesting older information published by Shiver and Henkart PA in Cell 1991 and 1992 indicating that RBL mast cells have some cytotoxic activity when transfected with perforin and Granzyme A, this may be worth to discuss.

Response: A note on the interaction between mast cells and EL4 lymphoma cells reported by Shiver and colleagues 1991 and 1992 has now been added (page 19).

Comments 4, 5, and 6: Line 66 some reference concerning mast cell promotion of angiogenesis is missing. Line 414 a reference concerning adenosine receptors seems to be missing. Line 192/223 typos FcepsilonRI.

Responses: Thank-you for bringing these to our attention. References of seminal studies on angiogenesis were incorporated in the text as well as appropriate references on mast cell expression of adenosine receptors. Typographical errors involving FcepsilonRI were also addressed.

Reviewer 2 Report

In this review, Hanes and collaborators have examined the controversial role of mast cells in skin and breast cancers. The authors introduced the role of mast cells and their mediators in allergic and non allergic disorders. Subsequently, they briefly discussed the controversial role of mast cells in different experimental and clinical cancers. The main subject of the review focuses on the role of mast cells in cutaneous and mammary tumors.

The paper is well written and covers several novel aspects of mast cell biology relevant in experimental and clinical skin and breast cancers.

Comments

  1. I would like to suggest slightly change the title of this review. Perhaps, a title like “Mast Cells and Skin and Breast Cancer: a Complex and Microenvironment-dependent Role.” better reflects the main topic of this review.
  2. Lines 181-184. There is ample evidence that several factors (e.g., low pH, etc.) in tumor microenvironment (TME) can profoundly alter mast cell activation. In addition, adenosine, discussed at lines 406-423, should be mentioned also here.
  3. Lines 186-188. There is overwhelming evidence that mast cells are a major source of several angiogenic factors. Incidentally, the Marshall group as well as other groups have contributed to this field. Perhaps, some of these papers should be cited.
  4. Lines 186-188. Human mast cells also produce lymphangiogenic factors (e.g., VEGF-C, VEGF-D), which are important in the formation of metastasis.
  5. Rodent and human mast cells synthesize cysteinyl leukotrienes (i.e., LTC4). There is evidence that cysteinyl leukotrienes are non-canonical angiogenic factors. This information should be mentioned.
  6. Figure 1. It is unclear whether the content of this figure refers to rodent and/or human mast cells. This point should be clarified. In addition, the production of lymphangiogenic factors should be included in this figure.
  7. Lines 500-508. The activating property of free light chain (FLCs) was reported long time ago and never confirmed. Perhaps, the authors can tone down this statement.

Author Response

In this review, Hanes and collaborators have examined the controversial role of mast cells in skin and breast cancers. The authors introduced the role of mast cells and their mediators in allergic and non allergic disorders. Subsequently, they briefly discussed the controversial role of mast cells in different experimental and clinical cancers. The main subject of the review focuses on the role of mast cells in cutaneous and mammary tumors.

The paper is well written and covers several novel aspects of mast cell biology relevant in experimental and clinical skin and breast cancers.

Comment 1: I would like to suggest slightly change the title of this review. Perhaps, a title like “Mast Cells and Skin and Breast Cancer: a Complex and Microenvironment-dependent Role.” better reflects the main topic of this review.

Response: We appreciate this suggestion. We also feel that this should be change coupled with replacement of “complex” with “complicated” in the title as per recommended by Reviewer #4 best represents our body of work.

Comment 2: Lines 181-184. There is ample evidence that several factors (e.g., low pH, etc.) in tumor microenvironment (TME) can profoundly alter mast cell activation. In addition, adenosine, discussed at lines 406-423, should be mentioned also here.

Response: Thank you for noticing this omission. Mention of acidity, adenosine, and free radicals have been added added to the introductory paragraph outlining mast cell key mediators (page 9).

Comment 3: Lines 186-188. There is overwhelming evidence that mast cells are a major source of several angiogenic factors. Incidentally, the Marshall group as well as other groups have contributed to this field. Perhaps, some of these papers should be cited.

Response: We agree with this comment and welcome the opportunity to reference this work.  References on angiogenesis have been added accordingly (page 9).

Comment 4: Lines 186-188. Human mast cells also produce lymphangiogenic factors (e.g., VEGF-C, VEGF-D), which are important in the formation of metastasis.

Response: As requested, we noted that mast cells can produce numerous VEGF isoforms and lymphangiogenic factors (page 9).

Comment 5: Rodent and human mast cells synthesize cysteinyl leukotrienes (i.e., LTC4). There is evidence that cysteinyl leukotrienes are non-canonical angiogenic factors. This information should be mentioned.

Response: This suggestion was appreciated. There’s certainly a role for many mast cell-derived lipid mediators in angiogenesis. Leukotrienes are now mentioned in the text and added to Figure 1 (page 9).

Comment 6: Figure 1. It is unclear whether the content of this figure refers to rodent and/or human mast cells. This point should be clarified. In addition, the production of lymphangiogenic factors should be included in this figure.

Response: Figure 1 serves as a general overview of how human mast cells may be differentially programmed. Though, much of the outlined functions for human mast cells are also pertinent to those of rodent origin.  We thank the reviewer for bringing the lack of clarity on this issue to our attention, as it would likely be a source of confusion. We have attempted to clarify in the modified figure 1 and have included mention of lymphangiogenic factors.

Comment 7: Lines 500-508. The activating property of free light chain (FLCs) was reported long time ago and never confirmed. Perhaps, the authors can tone down this statement.

Response. We agree this work is considered rather controversial, although there are not clear studies refuting these findings. We have softened the statement regarding the role of free light chains (page 21).

Reviewer 3 Report

This is a comprehensive and timely review to discuss the association between mast cell biology with tumorigenicity. The article is well written and I only have a minor comment on Figure-1 that: it is a little confusing to define the “anti-inflammatory mediators” there. The interleukins (ILs) listed in Figure-1 have been reported in literature as the pro-inflammatory cytokines to mediate the Th1 (such as IL-1 and IL-12) or Th2 (such as IL-4 and IL-5) inflammation. It would be nice to provide an explanation or discussion on the definition of the “anti-inflammatory mediators” in Figure-1.

Author Response

This is a comprehensive and timely review to discuss the association between mast cell biology with tumorigenicity. The article is well written and I only have a minor comment.

Comment 1: Figure-1: it is a little confusing to define the “anti-inflammatory mediators” there. The interleukins (ILs) listed in Figure-1 have been reported in literature as the pro-inflammatory cytokines to mediate the Th1 (such as IL-1 and IL-12) or Th2 (such as IL-4 and IL-5) inflammation. It would be nice to provide an explanation or discussion on the definition of the “anti-inflammatory mediators” in Figure-1.

Response: Point well-taken. IL-1 and IL-12 were moved to the “Pro-inflammatory” category. We acknowledge that many of the listed mediators play a multitude of roles depending on the context, making it difficult to label with a single function. We have added asterisks to IL-4 and -5 to highlight this dual function as pro- and anti-inflammatory mediators to make this fact more apparent.

Reviewer 4 Report

Mast Cells and Tumors: A Complex and Microenvironment-dependent Role
For the title, I suggest the author use this one.
Mast Cells and Tumors: A Complicated and Microenvironment-dependent Role
I also suggest the author use Complicated replace Complex in abstract.

Complex is used to refer to the level of components in a system. If a problem is complex, it means that it has many components. Complexity does not evoke difficulty.
On the other hand, complicated refers to a high level of difficulty. If a problem is complicated, there might be or might not be many parts but it will certainly take a lot of hard work to solve.
Complex means a system which is elegant, reasonable and beautiful but takes time to learn and comprehend. Complicated means a system which is ugly and cobbled together without any explainable justification other than 'it seems to work.
In the section of Key Mast Cell Mediators that Influence Cutaneous and Mammary Tumors Should be Mast Cell Key Mediators
Mast Cell Mediators: Their Differential Release and the Secretory Pathways Involved
In this part :TNF family members
TNF should all are TNF-alpha
Reference such as TNF-alpha is crucial for the development of mast cell-dependent colitis in mice
In Figure 1. TNF-alpha and INFy
Mast cells are important sentinel cells in host defense against infection and major effector cells in allergic disease.
When we talk about the Mast Cells and Tumors, we must mention IL-17.
As the author mentioned, We specifically address the key mediators produced by mast cells which impact tumor development.
So, I suggest the author add the part of IL-17 and Mast Cells. References such as:
The role of IL-17-secreting mast cells in inflammatory joint disease
Mast Cells Comprise the Major of Interleukin 17-Producing Cells and Predict a Poor Prognosis in Hepatocellular Carcinoma
In the section of Conclusions and the future landscape, should add more content.
I am happy to read the revised version of the manuscript.

Author Response

Comment 1: For the title, I suggest the author use this one.

Mast Cells and Tumors: A Complicated and Microenvironment-dependent Role

I also suggest the author use Complicated replace Complex in abstract.

Response: Excellent suggestion. We amended the title as per your recommendation while also addressing a related comment from another reviewer.

Comment 2: In the section of Key Mast Cell Mediators that Influence Cutaneous and Mammary Tumors Should be Mast Cell Key Mediators

Response: Corrected (page 8).

Comment 3: In this part :TNF family members. TNF should all are TNF-alpha. In Figure 1. TNF-alpha and INFy Reference such as TNF-alpha is crucial for the development of mast cell-dependent colitis in mice

Response: Thank you for these comments.  We recognize that different abbreviations still exist in literature for interferon-γ and tumor necrosis factor. However, the up to date nomenclature for TNF family members indicates that it should be referred to as TNF (Tumor necrosis factor superfamily members: nomenclature update and models (jax.org) and most current major immunology sources now use the abbreviation IFN in most cases for interferon.

Comment 4: Mast cells are important sentinel cells in host defense against infection and major effector cells in allergic disease. When we talk about the Mast Cells and Tumors, we must mention IL-17.

As the author mentioned, We specifically address the key mediators produced by mast cells which impact tumor development. So, I suggest the author add the part of IL-17 and Mast Cells. References such as:

The role of IL-17-secreting mast cells in inflammatory joint disease

Mast Cells Comprise the Major of Interleukin 17-Producing Cells and Predict a Poor Prognosis in Hepatocellular Carcinoma

Response: Thank you for this comment. IL-17A certainly influences skin and breast tumors. We have added a section to briefly highlight the roles for IL-17A in hosts with skin and breast tumors (page 12).

Comment 5: In the section of Conclusions and the future landscape, should add more content.

Response: Thank you for this suggestion. Further discussion on how technical advances may improve our understanding of mast cells have been added (page 24